# Toxic Potential of Cerrado Plants on Different Organisms

**DOI:** 10.3390/ijms23073413

**Published:** 2022-03-22

**Authors:** Jamira Dias Rocha, Fernanda Melo Carneiro, Amanda Silva Fernandes, Jéssyca Moreira Morais, Leonardo Luiz Borges, Lee Chen-Chen, Luciane Madureira de Almeida, Elisa Flávia Luiz Cardoso Bailão

**Affiliations:** 1Laboratório de Biotecnologia, Câmpus Central, Universidade Estadual de Goiás, Anápolis 75132-903, GO, Brazil; jamiradias@gmail.com (J.D.R.); jessycamoreiramorais@gmail.com (J.M.M.); leonardo.borges@ueg.br (L.L.B.); luciane.almeida@ueg.br (L.M.d.A.); 2Laboratório de Ficologia, Unidade Universitária de Goiânia-Laranjeiras, Universidade Estadual de Goiás, Goiânia 74863-250, GO, Brazil; fermelcar@gmail.com; 3Laboratório de Radiobiologia e Mutagênese, Departamento de Genética, Instituto de Ciências Biológicas I, Universidade Federal de Goiás, Goiânia 74045-155, GO, Brazil; fer.amanda7@gmail.com (A.S.F.); chenleego@yahoo.com.br (L.C.-C.)

**Keywords:** Brazilian savanna, chemical compounds, microorganisms, natural products, plant extract, tumor cells

## Abstract

Cerrado has many compounds that have been used as biopesticides, herbicides, medicines, and others due to their highly toxic potential. Thus, this review aims to present information about the toxicity of Cerrado plants. For this purpose, a review was performed using PubMed, Science Direct, and Web Of Science databases. After applying exclusion criteria, 187 articles published in the last 20 years were selected and analyzed. Detailed information about the extract preparation, part of the plant used, dose/concentration tested, model system, and employed assay was provided for different toxic activities described in the literature, namely cytotoxic, genotoxic, mutagenic, antibacterial, antifungal, antiviral, insecticidal, antiparasitic, and molluscicidal activities. In addition, the steps to execute research on plant toxicity and the more common methods employed were discussed. This review synthesized and organized the available research on the toxic effects of Cerrado plants, which could contribute to the future design of new environmentally safe products.

## 1. Introduction

Among the natural products found in plants, secondary metabolites are particularly important for humans [1]. These compounds exhibit different biological activities and have a wide range of uses. Secondary metabolites have been used as biopesticides, herbicides, cosmetics, and food additives, and have been used to improve human health significantly [1]. Secondary metabolites have been used in pharmaceutical product development, with approximately 50% of all drugs currently in clinical trials being derived from plants [2].

Although secondary metabolites are mainly used for beneficial biological activities, some are highly toxic [3]. The toxicity of a substance concerns its ability to cause harmful effects, which can be observed in a single cell, a group of cells, an organ system, or the entire body. Secondary metabolites can act by different mechanisms to exert toxic effects, making these natural compounds very useful in the pharmaceutical, agricultural, and food industries.

Identifying new natural compounds with specific toxicities is essential to reduce the use of synthetic chemicals that lead to increased resistance in pests or pathogens in both the agricultural and medical sectors. Drug discovery has developed significantly in recent decades but an urgent need remains for less toxic drugs with greater efficacy and economic accessibility. Plant-derived bioactive phytochemicals are promising novel compounds that could address some of these problems. Therefore, there is a continuous need to explore new active molecules with different mechanisms of action within the plant kingdom. Secondary metabolites in plants are defensive toxic compounds capable of inhibiting vital processes when touched and/or ingested. Phytochemical biomolecules can maximize the effectiveness and specificity of future drug design because they often have specific or multiple targets, and are both economically and ecologically sustainable [4,5].

The vast and unique biodiversity of the Cerrado biome contains many bioactive compounds [6], which enable Brazilian researchers to carry out sustainable research and to develop innovative products based on these compounds. The Brazilian Cerrado has 5% of the world’s biodiversity and 44% of the Brazilian flora [7,8,9]. This biome comprises a mosaic of various types of vegetation consisting of plant formations ranging from grassland, savanna, and even forest physiognomies, such as dry forests and gallery forests [10]. This diversity of environments influences the abundance of herbaceous, shrub, arboreal, and vine plants, consisting of more than 12,000 species that occur spontaneously in the Cerrado domain, with a high degree of endemicity [9,11,12]. The Fabaceae, Myrtaceae, Melastomataceae, Lauraceae, and Rubiaceae families are the most prominent in this biome regarding species richness [13]. The Cerrado flora is used by traditional populations (quilombolas, riverside dwellers, healers, and indigenous people). Various Cerrado plants, such as *Caryocar brasiliense*, *Mauritia flexuosa*, *Hancornia speciosa*, *Dypteryx alata*, and *Eugenia dysenterica*, are used ancestrally by local people as food and for therapeutic purposes in the treatment of various diseases [14,15]. It is important to highlight that the knowledge of these traditional populations associated with the use and application of natural products from the Cerrado contributes to the institution of this biome as a national heritage of great importance.

Cerrado plants have many secondary metabolites that act alone or synergistically to produce beneficial or harmful bioactivities depending on the point of view. For example, a toxic activity of a Cerrado biomolecule against insects could be beneficial to humans because we could use this valuable information to develop products to control disease vectors or agricultural plagues. Thus, in this review, we aimed to synthesize the information available about the toxicity of Cerrado plants, especially the secondary metabolites, on different organisms. This information provides the basis for future studies to develop novel bioactive compounds based on these plants for the control of human diseases and agricultural pests, and highlights the importance and fragility of this biome. Ongoing conservation of the Cerrado biome is vital for sustaining local communities and preserving endemic plant biodiversity.

## 2. Results and Discussion

### 2.1. Toxic Activity of Cerrado Plants

Although Cerrado plants are used in traditional medicine (Table 1), their biological activity is often not scientifically determined and their toxicity is unknown. Based on the literature search, the most common toxic qualities of Cerrado plants are antibacterial, antiparasitic, cytotoxic, insecticide, antifungal, and antiviral activities (Figure 1a and Appendix A). In total, 194 different plant species from the Cerrado biome with potential toxic activity were identified in this literature search (Appendix A). The species *Cochlospermum regium* (Bixaceae) was mentioned in most studies (*n* = 14) and had the following bioactivities: antibacterial, antifungal, cytotoxic, and mutagenic (Figure 2 and Figure 3). *C. regium* is a shrub widely distributed in Brazil and requires careful conservation based on the medicinal potential of its roots (Table 1). Since the harvesting of the roots kills the plant, it is in danger of being overexploited [16]. *E. dysenterica* (Myrtaceae) has the widest array of different bioactivities among plants included in the literature search, including antibacterial, antifungal, antiviral, cytotoxic, antiparasitic, molluscicide, and mutagenic activities (Figure 2 and Figure 3). *E. dysenterica* is native to the Cerrado and is highly regarded by local populations for its medicinal uses [15]. Different parts of this plant are used in traditional medicine to treat various disorders (Table 1). The wide distribution and popularity of these species contributed to the high number of studies on their bioactive compounds.

### 2.2. Toxic Cerrado Plant Families

Diverse plant families can cause toxicity on different cells or organisms (Appendix A). In the present review, we found 53 different plant families with toxic properties, the most represented of which were the Fabaceae and Myrtaceae families (Figure 1b). Fabaceae and Myrtaceae are the most frequently studied plant families in the Brazilian Cerrado and are also present in more than 80% of the localities sampled [13]. The large number of studies on these plant families may be due to their widespread occurrence, which means that they are easy to collect and more likely to be used as traditional medicine.

Some botanical families were significantly associated with bioactive properties (Figure 4). The Myristicaceae, Ericaceae, Polygonaceae, Vitaceae, and Ochnaceae families are associated with antiviral activity. The Siparunaceae, Phytolaccaceae, Euphorbiaceae, Aristolochiaceae, and Arecaceae are related to antibacterial activity. Nyctaginaceae is associated with antifungal activity. Sapindaceae, Malvaceae, Ebenaceae, and Solanaceae are associated with antiparasitic activity, while the Metteniusaceae family is associated with a molluscicidal activity. Piperaceae and Meliaceae are associated with insecticidal activity. Sapotaceae, Erythroxylaceae, Costaceae, Clusiaceae, Lythraceae, and Celastraceae are associated with cytotoxicity, predominantly against tumor cells (Appendix A).

Other than cytotoxicity against tumor cells, Cerrado plants had low genotoxicity, mutagenicity, and toxic effects in acute and chronic treatment regimens using murine models (Figure 4 and Appendix A). This low toxicity against mammals suggests that medicinal plants originating from the Brazilian Cerrado are generally safe to handle and could be used to develop safe and effective drugs, such as insecticides, antimicrobials, and antiparasitic drugs.

### 2.3. Experimental Design for Evaluating Plant Toxicity

The toxicity of plants is often complex and requires a careful experimental design to evaluate and characterize this toxic potential (Figure 5). First, it is necessary to choose the target plant species and the more appropriate part of the plant. Various approaches have been proposed, including (i) random selection based on plant availability, (ii) chemotaxonomic or phylogenetic selection based on known chemical classes in a particular genus or species, and (iii) ethnopharmacological selection based on the prior use of a particular plant in local or traditional medical practice [125]. In the present review, most studies focused on plants’ leaves, roots, and stems rather than fruit or seeds (Figure 1c). Secondary metabolites vary depending on the part of the plant consumed, with different amounts of specific secondary metabolites accumulating in different plant parts [126]. From a conservation perspective, it should be noted that the collection of root specimens usually leads to the death of the plant.

After selecting the plant species, it is crucial to choose the collection site by considering the environmental factors that affect the production of secondary metabolites, such as season, circadian rhythm, temperature, altitude, atmospheric composition, soil fertility, humidity, solar radiation, wind, herbivory, air pollution, and soil pollution [126,127]. After collecting the plant samples, the correct identification of the species should preferably be carried out by a botanist and an exsiccate must be deposited into an herbarium [125].

Quality control and standardization of all processing stages are fundamental to the successful characterization of plant-derived bioactive compounds. These steps ensure the reproducibility and safety of plant-derived products [15]. Therefore, the collected material must be dried with air circulation and stored in low humidity and temperature. Grinding should only be performed when preparing the extracts. Extracts are usually prepared by percolation (cold extraction method is commonly used), by a Soxhlet extractor (hot extraction method), or by an acid base. A polar solvent (methanol or ethanol) is generally used for single extractions (cold or hot). For multiple extractions, three types of solvents are usually used: non-polar (hexane or petroleum ether), moderate polarity (chloroform or dichloromethane), and polar (methanol or ethanol) [125]. However, it is important to highlight that organic solvents are often toxic and reuse is not always possible. As a result, great efforts are being made to replace conventional organic solvents with less toxic solvents, such as supercritical fluids, ionic liquids at room temperature, perfluorinated hydrocarbons, and water, to decrease the release of toxic solvents into the water, air, and soil, and thus to reduce the amount of environmental pollution [128]. In the present review, most studies (31.85%) used ethanol as the extraction solvent (Figure 1d). Ethanol is a suitable solvent for polyphenol extraction and is considered safe for both human and environmental health [129].

In general, the liquid extract obtained must be concentrated. Once the concentrated extract is obtained, several quality parameters are essential for standardization, such as pH, solid content, density, chemical marker content, and viscosity. After considering the chemical and physical stability of the chemical extract, drying is the most commonly used preservation method to obtain a stable plant product [15]. At this point, the investigation into the chemical constituents and/or toxic activities of the plant material can begin (Figure 5).

The regulatory compliance of toxicity assessments is mainly handled globally by the Organization for Economic Cooperation and Development (OECD). Until recently, toxicological analyses were primarily performed using animal models. However, in vitro and in silico analyses are becoming more acceptable in regulatory settings as an alternative to animal testing [3,130], which can reduce the cost and duration of these tests, as well as reduce the number of experimental animals used [130]. Different toxic prediction tools have become more accurate and effective [130,131]. The “-omics era” (concerning genomic, transcriptomic, proteomic, and metabolomic data) has enabled researchers to derive hypotheses on the mechanisms of action and target identification of chemical compounds using high-throughput specialized instrumentation. These techniques offer whole-organism data rather than specific information on a particular pathway or target [132]. However, bioactive promiscuity, lack of complete genome sequence data, poor gene annotation, high costs, expensive and specific equipment, and the need for qualified, trained personnel remain as limiting factors in the use of omics technology in this field.

Different testing systems exist to determine if a substance is toxic and many different toxic endpoints may be considered such as cytogenotoxicity, carcinogenesis, hepatotoxicity, renal toxicity, neurotoxicity, reproductive toxicity, endocrine toxicity, and immunotoxicity [133]. Toxicity assessments are essential for developing drugs, agrochemicals, cosmetics, food additives, and other important products.

The cytotoxic activity of plant extracts or isolated compounds can be determined using methods that evaluate (i) cell morphology variations using fresh cell preparations; (ii) cell membrane integrity using dye exclusion assays such as trypan blue and Congo red; and (iii) the inhibition of cellular metabolism using MTT and resazurin reduction assays, which evaluate the mitochondrial function of cells by measuring the activity of mitochondrial enzymes [125,134]. In the present review, the most commonly used method for determining the cytotoxic potential of Cerrado plants was the MTT assay (Figure 1f). This method to determine cytotoxicity and cell viability is easy to use, safe, and has high reproducibility [134].

A variety of laboratory methods can be used to evaluate or screen the in vitro antimicrobial activity of an extract or pure compound. The most well-known and simple methods to detect antibacterial and antifungal compounds are disk diffusion and broth or agar dilution methods. More sophisticated techniques, such as flow cytofluorometric and bioluminescent methods, can be employed but they are not widely used because they require specific and expensive equipment [135]. In the present review, the broth microdilution assay was the most commonly used method to determine the antibacterial and antifungal properties of Cerrado plants (Figure 1f). Dilution methods are appropriate for determining the minimum inhibitory concentration (MIC) of a compound or extract, which is the lowest concentration of an antimicrobial that inhibits visible growth [135]. The methods commonly used for in vitro evaluation of antiviral activity are based on the ability of viruses to replicate in cultured cells because they are obligate intracellular symbiotes. Some viruses cause cytopathic effects or form plaques. Others can produce specialized functions or cell transformations. Viral replication can also be monitored by detecting viral products, such as viral DNA, RNA, or polypeptides [136]. The cytopathic effect inhibition assay is one of the most reliable and robust assays for screening natural antiviral compounds [137]; is a rapid and sensitive method; and has been extensively used to detect the antiviral potential of Cerrado plants (Figure 1f).

Unlike assays used to determine the antibacterial, antifungal, and antiviral activity of plant products, bioassays for parasites tend to be highly species-specific [136]. To improve the performance of antiparasitic assays, the following should be carefully considered: (i) the use of a well-characterized, drug-sensitive parasite strain, with validated model availability, which is safe for the researcher, and (ii) the use of sensitive endpoint-reading techniques [136]. The cytotoxic potential of natural products on *Leishmania* spp. and *Trypanosoma* spp. was evaluated by the MTT assay, which was widely used in the articles included in the present review (Figure 1f).

Similar to antiparasitic assays, bioassays for substances that control insects are highly variable due to the abundance and variety of insects and their life cycle stages [138]. Notably, the insects used in the assay should have been standardized concerning species, age, and physiological state [138]. In general, topical application is used to study the insecticide potential of natural products because it has a faster response than ingestion and is independent of insect activity. The disadvantages of topical application are that the compound may not overcome penetration barriers, the application process is tedious, and the process requires manual dexterity [139,140]. Tests on larvae are preferred because insecticides that are effective on larval stages can prevent the development of the next generation of insects [140]. Bioassays performed under conditions that simulate management applications are also required; however, formulated products should be used to ensure standardization. On-host applications or field tests should be considered but present a particular challenge because of the possible interactions with the host [139]. The larvicidal activity assay is one of the most commonly used assays when studying natural compounds with insecticide potential (Figure 1f). However, topical tests are scarce in Cerrado plants.

### 2.4. Toxicity of Secondary Metabolites

Secondary metabolites are organic molecules that are not involved in the normal growth and development of an organism. The absence of secondary metabolites does not result in immediate death but rather in a long-term impairment of the organism’s survivability, as they often play an essential role in plant defense. Toxicity is, therefore, an excellent strategy to inhibit the action of predators. Secondary metabolites act on the predators through multiple mechanisms (Figure 6). They can interact specifically or not specifically with proteins (enzymes, receptors, ion channels, and structural proteins), nucleic acids, biomembranes, and other cellular components [141,142]. The interaction with these different targets can disturb the vital components of the cellular-signaling system, resulting in dysregulated essential signaling in the nervous system (e.g., concerning neurotransmitter synthesis, storage, release, binding, re-uptake, receptor activation and function, and enzymes involved in signal transduction) or in interference with vital enzymes and blocking of metabolic pathways [143]. When interacting with nucleic acids, some secondary metabolites can have both mutagenic and antimutagenic roles, and act as a mutagen by directly binding to DNA, generating ROS, or inhibiting topoisomerase enzymes [144].

Secondary metabolites can be simply classified into three main groups: (i) terpenes (such as plant volatiles, cardiac glycosides, carotenoids, and sterols); (ii) phenolics (such as phenolic acids, coumarins, lignans, stilbenes, flavonoids, tannins, and lignin); and (iii) nitrogen-containing compounds (such as alkaloids and glucosinolates) [145]. In the present review, 60 compounds with toxic activity were detected among the studied plants (Appendix A and Appendix A). The most representative secondary metabolites isolated from Cerrado plants with toxic activities were terpenes, flavonoids, and alkaloids (Figure 1e). Many alkaloids are toxic and can cause death, even in small quantities. It seems that alkaloid function in plants and animals is linked to defense mechanisms, including antibiotic activities [145]. The beneficial antibiotic effects of plant secondary metabolites could therefore be similarly useful in human medical interventions, although care should be taken to establish safety profiles for plant-derived extracts.

## 3. Materials and Methods

The review was performed using the PubMed (*n* = 314), Science Direct (*n* = 2184), and Web of Science (*n* = 378) databases. In total, 2876 abstracts were selected using the following search terms: “Cerrado” AND “cytotoxic*” OR “genotoxic*” OR “insecticide*” OR “antiparasitic*” OR “antibacterial*” OR “antifungal*” OR “molluscicide*” OR “antiviral*” OR “chronic toxicity*” OR “acute toxicity*” OR “mutagenic*”. The asterisk (*) was used as a wildcard and enabled the search of any letters in its place. The inclusion criteria were species (i) native to the Cerrado biome and (ii) presenting toxic activity. Gray literature and review articles were excluded (PubMed (*n* = 93), Science Direct (*n* = 1963), and Web of Science (*n* = 157)). Studies that overlapped were also excluded (*n* = 34). Thus, 2665 articles were considered to be outside the scope of this review and were excluded. A total of 187 articles published between 2000 (first record within the inclusion and exclusion criteria) and December 2020 were selected and analyzed (Figure 7).

We extracted the species, part of the plant, type of extract, dose/concentration, activity, and extraction method used from each manuscript in our analysis. The plant species were then classified into their respective families according to the Flora do Brasil website [22]. The frequency of each type of toxic activity reported was associated with plant families by generating a heatmap in R [146] using the “pheatmap” package [147].

## 4. Conclusions

The present review summarizes the literature from the last two decades related to the toxicity of plant species from the Cerrado biome and the secondary metabolites that have been both identified and evaluated for their toxicity. The species and compounds reported in the present review have high cytotoxicity against tumor cells and low toxicity against non-tumor cells, indicating that Cerrado plants could be used to develop new anti-cancer drugs. Plants from the Cerrado biome presented low genotoxicity, mutagenicity, and toxic effects on murine models in acute and chronic treatments. Moreover, Cerrado plants are effective against bacteria, fungi, viruses, insects, and parasites. In combination, these data suggest that Cerrado plants can be used to develop products that can be safely handled and administered (because of the low toxicity on mammals), including insecticides against urban and agricultural pests, antimicrobials, and antiparasitic products. The notable limitations of this review are the relatively low number of studies investigating the molluscicidal activity and the scarcity of associated omics data. We hope that this review supports the conservation of the Cerrado biome against anthropogenic activities, ensuring the preservation of the vast biodiversity and natural wealth provided by this unique biome.

## Figures and Tables

**Figure 1 ijms-23-03413-f001:**
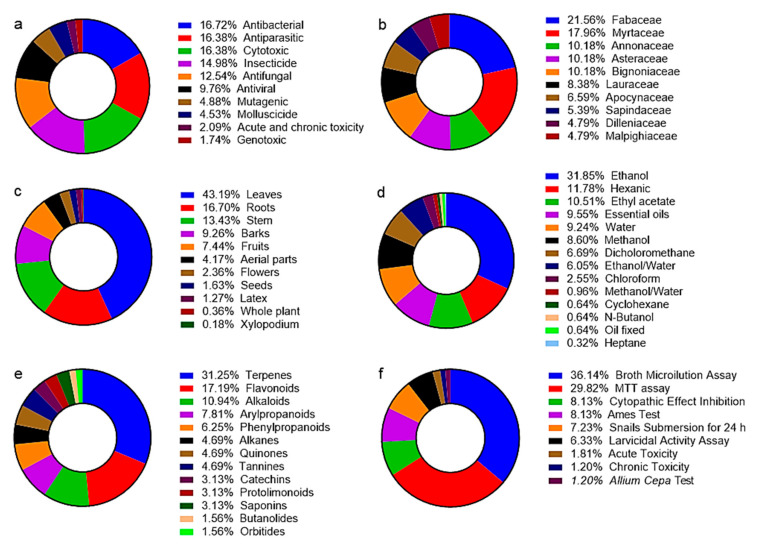
Summary of studies on the toxic activities of Cerrado plants included in the present review. The included manuscripts were screened to generate donut charts to visualize the proportions of (**a**) toxic activities studied, (**b**) plant families studied, (**c**) part of the plant studied, (**d**) type of extract or fraction studied, (**e**) classes of secondary metabolites studied, and (**f**) main techniques used to assess the toxicity of medicinal plants.

**Figure 2 ijms-23-03413-f002:**
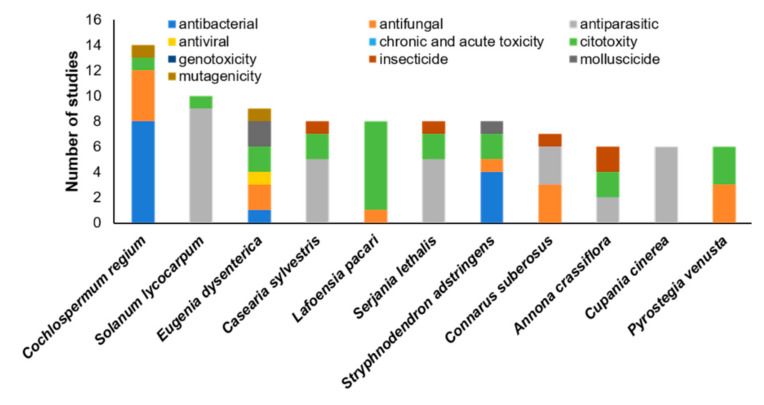
The bioactive properties of the Cerrado plant species that have been investigated in multiple studies. The most studied Cerrado species was *Cochlospermum regium*, while *Eugenia dysenterica* had the most diverse bioactive properties.

**Figure 3 ijms-23-03413-f003:**
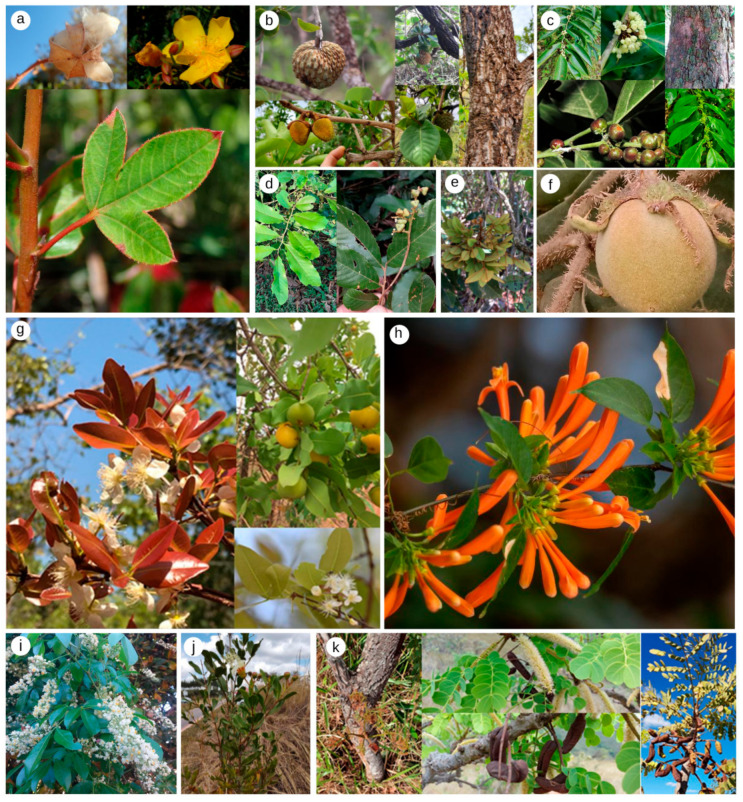
Most representative Cerrado species with toxic activity on different organisms according to this literature survey. (**a**) *Cochlospermum regium* (Mart. ex Schrank) Pilg. (“algodãozinho-do-campo”); (**b**) *Annona crassiflora* Mart (“araticum”); (**c**) *Cupania cinerea* Poepp. and Endl; (**d**) *Casearia sylvestris* Sw. var. sylvestris *(*“guaçatonga”); (**e**) *Connarus suberosus* Planch (“bico de papagaio”); (**f**) *Solanum lycocarpum* A.St.-Hil. (“lobeira”); (**g**) *Eugenia dysenterica* (Mart.) DC (“cagaita”); (**h**) *Pyrostegia venusta* (Ker Gawl.) Miers (“cipó-de-são-joão”); (**i**) *Serjania lethalis* A.St.-Hil. (“cipó-timbó”); (**j**) *Lafoensia pacari* A.St.-Hil. (“pacari”); and (**k**) *Stryphnodendron adstringens* (Mart.) Coville (“barbatimão”). All photographs were obtained from the Herbário da Universidade Estadual de Goiás (HUEG) and are available at https://www.gbif.org/pt/dataset/bbb1f181-3221-4a10-ad52-14f1da0dca26 (accessed on 23 October 2021).

**Figure 4 ijms-23-03413-f004:**
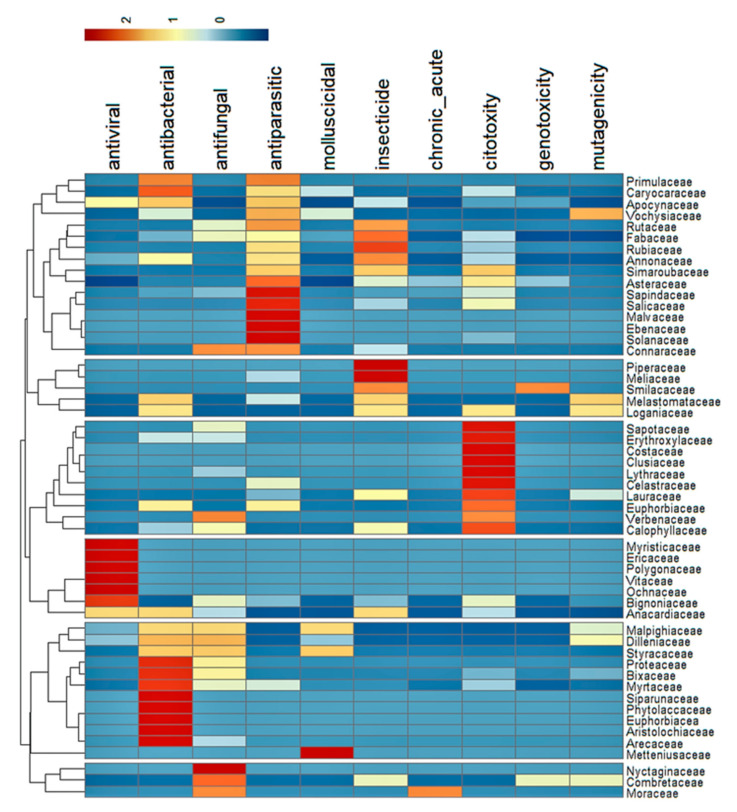
Heatmap of the plant families included in the present review grouped according to the frequency of the important bioactive properties associated with each family.

**Figure 5 ijms-23-03413-f005:**
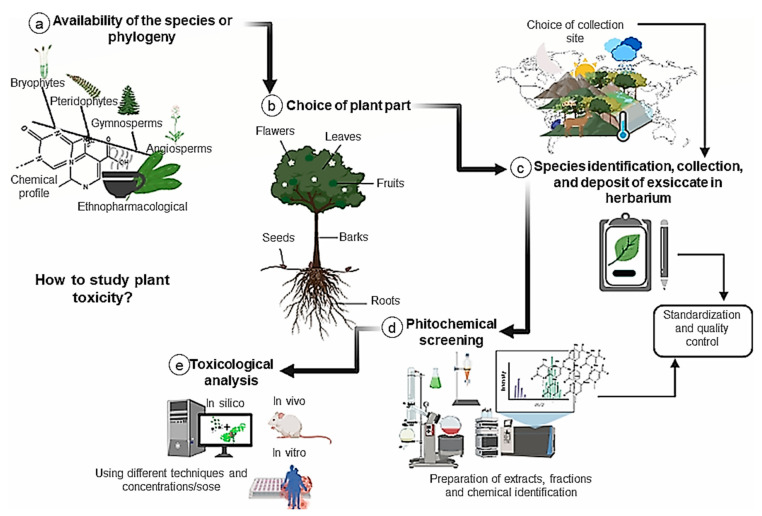
Proposed workflow for the effective study of plant toxicity. The study of plant toxicity should be carefully designed with the following steps carefully considered: (**a**) Selection of species according to plant availability, chemotaxonomy/phylogenetics, or ethnopharmacology. (**b**) Selection of the part of the plant to be used. It is important to understand that environmental factors also affect the production of secondary metabolites in different parts of the plant. (**c**) Identification of species, collection, and deposition of the exsiccate into an herbarium. (**d**) Obtainment of extracts by percolation, Soxhlet extractor, or acid-base strategies. Various quality parameters are used to standardize the preparation of samples (pH, solids content, density, content of chemical markers, and viscosity). At this stage, it is common to investigate the chemical constituents of the extract. (**e**) Toxicological analysis of the plant material using different experimental methods (in silico, in vitro, and/or in vivo).

**Figure 6 ijms-23-03413-f006:**
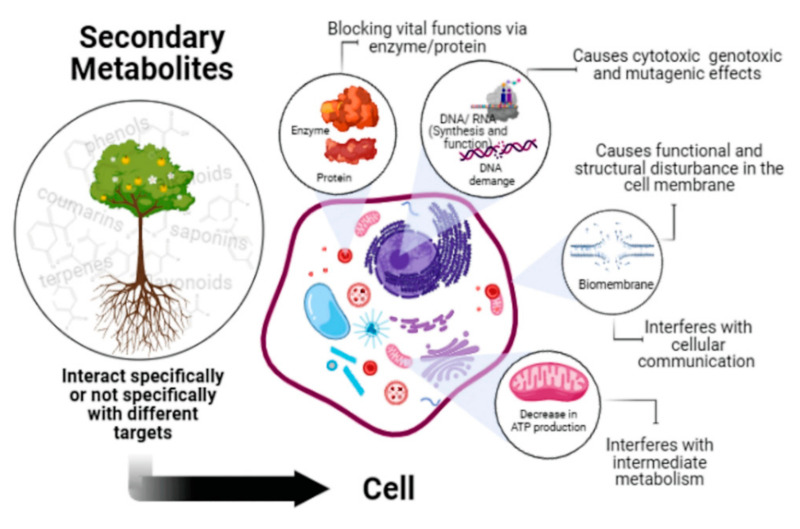
Mechanisms of action of secondary metabolites with cytotoxic effects. Secondary metabolites can interact specifically or not specifically with biomolecules, biomembranes, and other cellular components, disturbing the vital components of the cell.

**Figure 7 ijms-23-03413-f007:**
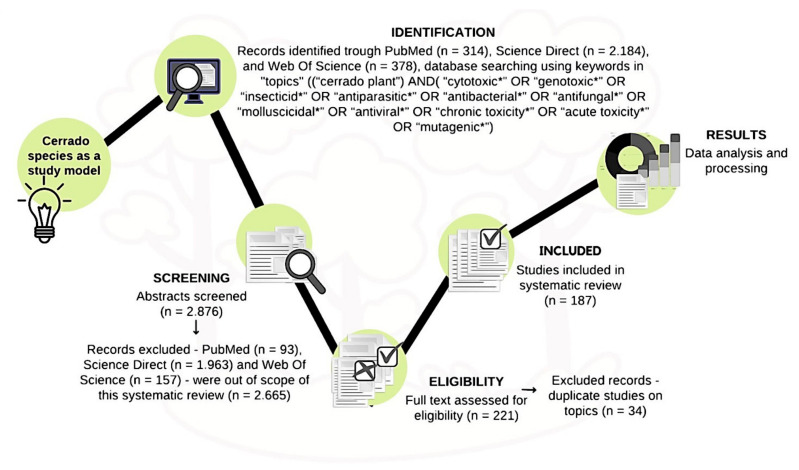
The experimental workflow used in the present review to identify articles containing information about Cerrado plants with toxic bioactivities. The workflow involved the identification, screening, eligibility assessment, and inclusion of available manuscripts from several online databases. During the search for the terms in the databases, the asterisk (*) was used as a wildcard and enabled the search of any letters in its place.

**Table 1 ijms-23-03413-t001:** Ethnobotanical data for the Cerrado biome plant species included in the present review.

Family/Scientific Name	Popular Name	Popular Use	Reference
**Anacardiaceae**			
*Anacardium occidentale* L.	Caju	Treatment of malaria and yellow fever	[17]
*Astronium urundeuva* (M.Allemão) Engl.or *Myracrodruon urundeuva* Allemão	Aroeira	Antiseptic for external ulcers	[18]
*Schinus terebinthifolius* var. *radiannus* Engl.	Aroeira-de-brejo and aroeira-da-praia	Treatment of leprosy and tumors	[19]
**Annonaceae**			
*Anaxagorea dolichocarpa* Sprague & Sandwith	Bananinha	Treatment of grippe and cold	[20]
*Annona coriacea* Mart.	Aaraticum	Treatment of dermatitis, and used as a depurative agent	[21]
*Annona crassiflora* Mart.	Araticum or marolo	Treatment of chronic diarrhea	[19]
*Annona mucosa* Jacq.	Araticum, Graviola Brava, Condessa, Fruta de Conde, Biribá, Fruta de Condessa, Fructa da Komdessa	N/F	[22]
*Cardiopetalum calophyllum* Schltdl.	Imbirinha	N/F	[22]
*Duguetia furfuracea* (A. St. Hil.) Benth & Hook	Araticum do cerrado or ata brava	Treatment of rheumatism and renal colic, and used as antihyperlipidemic and anorexic agent	[23]
*Duguetia lanceolata* A.St.-Hil.	Pindaíba, Pindahiba, Pindaúba, Capreuva Vermelho	N/F	[22]
*Xylopia aromatica* (Lam.) Mart.	Pimenteira	Treatment of digestive problems and inflammation, and used as tonic and aphrodisiac	[24]
*Xylopia emarginata* Mart.	Pindaíba-do-brejo	N/F	[25]
**Apocynaceae**			
*Aspidosperma macrocarpon* Mart. & Zucc.	Peroba-gigantedo-cerrado	Antimalaric and anti-inflammatory	[26]
*Aspidosperma tomentosum* Mart.	Guatambu	Treatment of gastritis	[27]
*Hancornia speciosa* Gomes	Mangaba, Mangabeira	Treatment of gastrointestinal diseases, tuberculosis, diabetes, hypertension, dermatitis, diarrhea, ulcers, gastritis, acne, warts, and cancer, and used as anti-inflammatory	[28,29]
*Himatanthus drasticus* (Mart.) Plumel	Janaúba and Tiborna	Treatment of cancer	[30]
*Himatanthus obovatus* (Müll. Arg.) Woodson	Angelica	Treatment of anemia, wound healing, cholesterol, pain, nose bleeding, hypertension, uterine inflammation, labyrinthitis, pneumonia, worms, and vitiligo, and is a blood cleanser and muscular relaxant	[27]
*Secondatia floribunda* A.DC.	Catuaba-de-rama or Catuaba-decipó	Treatment of sexual impotence, nerve complications, depression, rheumatism, and inflammatory conditions	[31]
**Arecaceae**			
*Attalea phalerata* Mart. ex Spreng.	Bacuri	Pulmonary decongestant, anti-inflammatory for joints, and is antipyretic	[32]
*Attalea speciosa*	N/F	N/F	
*Mauritia flexuosa* L.f.	Buriti	Treatment of burns and used as a potent vermifuge	[22]
**Aristolochiaceae**			
*Aristolochia cymbifera* Mart. & Zucc	Caçaú, milhome, Crista-De-Galo	Treatment of oral diseases	[33,34]
**Asteraceae**			
*Ageratum conyzoides* L.	Mentrasto	Treatment of malaria, ulcers, dysentery, and yellow fever, and is a purgative, febrifuge, anti-microbial, and anti-lytic agent	[35]
*Ageratum fastigiatum* (Gardner) R.M.King & H.Rob.	Mata pasto	Cicatrizing and anti-inflammatory, and is an analgesic and antimicrobial agent	[36]
*Aldama discolor* (Baker) E.E.Schill. & Panero	N/F	N/F	[22]
*Baccharis dracunculifolia* DC.	Alecrim-do-campo and vassourinha	Anti-inflammatory agent mainly for the treatment of gastrointestinal diseases	[37]
*Chromolaena squalida* (DC.) R.M.King & H.Rob.	N/F	N/F	[22]
*Cyrtocymura scorpioides* (Lam.) H.Rob.	Piracá, Enxuga or Erva-de-São-Simão	Treatment of dermal diseases, including chronic wounds and ulcers	[38]
*Eremanthus incanus* (Less.) Less.	N/F	N/F	[22]
*Lychnophora pinaster* Mart.	Arnica	Treatment of inflammation, pain, rheumatism, contusions, bruises, and insect bites	[39]
*Lychnophora trichocarpha* Spreng.	Arnica	Treatment of inflammation and rheumatologic diseases, and is an insecticide agent	[39]
*Mikania laevigata* Sch.Bip. ex Baker	Guaco	Treatment of inflammatory disorders, such as bronchitis, chronic lung diseases, and bronchial asthma	[40]
*Piptocarpha rotundifolia* (Less.) Baker	N/F	N/F	[22]
*Pseudogynoxys cabrerae* H.Rob. & Cuatrec.	N/F	N/F	[22]
*Vernonanthura polyanthes* (Spreng.) A.J. Vega & Dematt.	Assa-peixe	Treatment of bronchitis, coughing, bruises, ocular inflammation, rheumatism, hemorrhoids, kidney disorders, and uterine infections	[41]
**Bignoniaceae**			
*Adenocalymma nodosum* (Silva Manso) L.G.Lohmann	N/F	N/F	[22]
*Amphilophium elongatum* (Vahl) L.G.Lohmann	N/F	N/F	[22]
*Anemopaegma setilobum* A.H. Gentry	N/F	N/F	[22]
*Arrabidaea brachypoda* (DC.) Bureau	Cipó-una, tintureiro or cervejinha do campo	Treatment of kidney diseases and painful joints (arthritis)	[42]
*Callichlamys latifolia* (Rich.) K. Schum.	Cipó-guachana amarelo	Treatment of intestinal colic and skin conditions	[43]
*Cuspidaria sceptrum* (Cham.) L.G.Lohmann	Lírio-do-campo	N/F	[22]
*Cybistax antisyphilitica* (Mart.) Mart.	Ipeˆ-branco, cincofolhas and pe’-de-anta	Depurative, antisyphilitic, and diuretic agents	[44]
*Distictella elongata* (Vahl) Urb.	N/F	N/F	[22]
*Fridericia chica* (Bonpl.) L.G.Lohmann	Carajuru or guajuru-piranga or Crajiru	Wound healing	[45]
*Fridericia craterophora* (DC.) L.G.Lohmann	Cipó-una, tintureiro or cervejinha do campo	Treatment of kidney diseases	[43]
*Fridericia formosa* (Bureau) L.G.Lohmann	N/F	N/F	[22]
*Fridericia platyphylla* (Cham.) L.G.Lohmann	Cipó-una, tintureiro or cervejinha do campo	Treatment of kidney diseases	[46]
*Fridericia samydoides* (Cham.) L.G.Lohmann	N/F	N/F	[22]
*Jacaranda cuspidifolia* Mart.	Jacarandá, caroba, caiuá, caroba-branca, pau-de-colher, dacarandá-de-minas	Treatment of syphilis and gonorrhea, and is an antimycobacterial activity	[47]
*Pyrostegia venusta* (Ker Gawl.) Miers	Cipó-de-são-joão	General tonic and used to treat diarrhea, vitiligo, and coughing	[48]
*Zeyheria tuberculosa* (Vell.) Bureau ex Verl.	Ipê Felpudo	Treatment of cancer and dermatosis	[49]
**Bixaceae**			
*Cochlospermum regium* (Mart. ex Schrank) Pilg.	Algodãozinho-do-campo, algodãozinho-do-cerrado, algodãobravo, periquiteira, algodão-do-mato, algodãozinho, algodãozinhocravo, algodoeiro-do-campo, butua-de-corvo, periquiteira-do-campo, pacote, ruibarbo-do-campo and sumaúma-do-igapó	Treatment of ulcers, arthritis, intestinal infections, gynecological infections, and skin diseases	[50]
**Calophyllaceae**			
*Calophyllum brasiliense* Cambess.	Guanandi, olandi, and jacareúba	Anti-inflammatory, used for treatment of rheumatism, vein-related problems, hemorrhoids, gastric ulcers, pain, inflammation, diabetes, hypertension, and herpes	[51]
*Kielmeyera coriacea* Mart. & Zucc.	Pau-Santo	Antiparasitic, antifungal, antibacterial, and antimalaria, used for treatment of schistosomiasis and leishmaniosis	[52]
*Kielmeyera lathrophyton* Saddi	Murici-pequeno	Treatment of schistosomiasis, leishmaniasis, malaria, and both fungal and bacterial infections	[53]
*Caryocar brasiliense* Cambess.	Pequi	Anti-inflammatory and used for treatment of high blood pressure	[54]
*Caryocar coriaceum* Wittm.	“pequi”, “piqui”, “pequá”, “Thorn almond”, “horse bean” or “Brazilian almond	Anti-inflammatory and used to promote healing	[55]
**Celastraceae**			
*Cheiloclinium cognatum* (Miers) A.C.Sm.	Bacuparí, pitombinha	Treatment of fever and edema	[56]
*Salacia crassifolia* (Mart. ex Schult.) G. Don	Bacupari, cascudo, and saputá	Treatment of pediculosis, kidney disease, gastric ulcers, skin cancer, malaria, chronic coughs, and headaches	[57]
**Clusiaceae**			
*Garcinia gardneriana* (Planch. & Triana) Zappi	Bacupari	Treatment of inflammation, pain, urinary infections, and other infections	[58]
**Combretaceae**			
*Terminalia argentea* Mart. & Zucc.	Capitão, capitão-do-campo or pau-de-bicho	Treatment of gastric ulcers, bronchitis and hemorrhages, ulcers, flu with fever, diarrhea, inflammation, wounds, cramps, cancer, rheumatism, and body pains, and used as tranquilizer, diuretic, and anti-anxiety agent	[59]
*Terminalia fagifolia* Mart.	Mirindiba, capitão do mato, capitão, capitão-do-cerrado and cachaporra do gentio	Treatment of oral mucosa lesions by *Candida* strains, tumors (breast cancer), and diseases of the gastrointestinal tract (diarrhea and gastritis)	[60]
**Connaraceae**			
*Connarus suberosus* Planch	Tropeiro or bico de papagaio, galinha-choca	Treatment of diarrhea and heart problems	[61]
**Costaceae**			
*Chamaecostus subsessilis* (Nees & Mart.) C.D.Specht & D.W.Stev.	N/F	N/F	[22]
*Costus spiralis* (Jacq.) Roscoe	Cana-de-macaco or cana-do-brejo	Treatment of urinary infections and kidney stones	[62]
**Dilleniaceae**			
*Curatella americana* L.	Lixeira cajueiro-bravo	N/F	[63]
*Davilla elliptica* A.St.-Hil.	Lixinha	Astringent tonic and purgative, used for treatment of swellings, especially of the lymphatic nodes and testicles	[64]
*Davilla nitida* (Vahl) Kubitzki	Cipó-de-fogo, sambaibinha, lixeirinha de rama	Treatment of gastric problems	[64]
*Davilla rugosa* Poir	Sambaibinha, Cipó de Carijó, Cipó-caboclo,	Treatment of ulcers	[18]
**Ebenaceae**			
*Diospyros hispida* A. DC.	Olho-de-boi	Treatment of pain and leprosy	[27]
*Diospyros lasiocalyx* (Mart.) B.Walln.	Olho-de-boi	Treatment of pain and leprosy	[27]
**Ericaceae**			
*Gaylussacia brasiliensis* Meisn	Camarinha	Treatment of inflammation	[65]
**Erythroxylaceae**			
*Erythroxylum daphnites* Mart.	Chapadinho, fruta-de-tucano, mercúrio and pimenta	N/F	[22]
*Erythroxylum subrotundum* A.St.-Hil.	N/F	N/F	[22]
*Erythroxylum suberosum* St. Hil.	Cabelo de negro	Abortive and used for prevention of inflammatory processes	[66]
**Euphorbiaceae**			
*Alchornea triplinervia* (Spreng.) Müll.Arg.	Tapiá	Treatment of gastric disturbances	[67]
*Croton heliotropiifolius* Kunth	velame	Treatment of influenza, general pain, inflammation, dermatitis, gastrointestinal disturbances, malaise, poor digestion, boils, and back pain, and used as a depurative agent	[21]
*Croton urucurana* Baill.	Sangra-d’agua	Treatment of cancer, prostate cancer, diabetes, stomach pain, gastritis, uterine inflammation, kidneys, and ulcers	[27]
*Croton velutinus* Baill.	Pimentinha	Treatment of cancer	[68]
**Fabaceae**			
*Anadenanthera colubrina* (Vell.) Brenan	Angico	Treatment of inflammation, respiratory problems related to infection (cough, influenza, and bronchitis), diarrhea, and toothache	[69]
*Bauhinia holophylla* (Bong.) Steud.	Pata-de-vaca	Treatment of diabetes and infections, and used as an analgesic, antidiarrheal, anti-inflammatory, and diuretic agent	[70]
*Bowdichia virgilioides* Kunth	Sucupira preta	Treatment of spinal pain, rheumatism, sexual impotence, bone pain, inflammation of the skin, general inflammation, inflammation of the uterus, wounds, general pain, back pain, vaginal inflammation, and throat pain, and used as a purifying agent	[47]
*Copaifera langsdorffii* Desf.	Copaíba	Anti-rheumatic, anti-inflammatory, and emollient agent; used as a general tonic; and used for treatment of wounds and infections of the bladder, inflammation, stomach aches, and uterine inflammation	[71]
*Copaifera multijuga* Hayne	Copaiba	Anti-rheumatic, anti-inflammatory, and emollient agent; used as a general tonic; and used for treatment of wounds and infections of the bladder, inflammation, stomach aches, and uterine inflammation	[72]
*Dimorphandra mollis* Benth.	Faveiro-de-anta	Treatment of inflammation (swelling/pain)	[56]
*Dipteryx alata* Vogel	Cumbaru	Treatment of dysentery, pain, throat pain, flu, snakebites, and coughs	[27]
*Enterolobium gummiferum* (Mart.) J.F.Macbr.	N/F	N/F	[22]
*Eriosema crinitum* (Kunth) G. Don	Pustemeira	Treatment of inflammatory diseases, including inflammatory skin disorders such as psoriasis	[73]
*Hymenaea courbaril* L.	Jatobá ORFarinheira	Treatment of diarrhea, dysentery, intestinal colic, pulmonary weakness, and chronic cystitis	[63]
*Hymenaea martiana* Hayne	Jatoba-da-mata	Treatment of gastrointestinal, urinary, and respiratory tract infections, as well as for inflammatory disorders (rheumatoid arthritis), liver problems, respiratory disorders, inflammation, and stomach and chest aches	[74]
*Hymenaea stigonocarpa* Mart. ex Hayne	Jatobá-do-cerrado	Treatment of diarrhea, infections, prostate cancer, anemia, leukemia, anxiety (tranquilizer), weakness, cataracts, eye irritation, asthma, bronchitis, flu, pneumonia, gastritis, indigestion, ulcers, inflammation, rheumatism, uterine and ovary infections, prostate diseases, kidneys, wounds, bone fractures, body pain, throat infections, throat inflammation, coughing with catarrh, and vomiting, and used as a depurative, expectorant, female intimate-cleaning, and lung-strengthening agent, and general tonic	[59]
*Inga laurina* (Sw.) Willd.	Ingá Branco	Anti-inflammatory and antidiarrheal, nasal decongestant, used for treatment of skin conditions and earaches, and for cleaning teeth	[75]
*Lachesiodendron viridiflorum* (Kunth) P.G. Ribeiro, L.P. Queiroz & Luckow	Surucucu	N/F	[22]
*Peltophorum dubium* (Spreng.) Taub.	N/F	N/F	[22]
*Plathymenia reticulata* Benth.	Candeia, vinhático	Treatment of hemorrhaging, swelling of injuries, liver, kidneys, and wounds	[76]
*Pterodon emarginatus* Vogel	Sucupira and sucupira-branca	Anti-inflammatory and analgesic agent	[56]
*Stryphnodendron adstringens* (Mart.) Coville	Barbatimão, casca-da-virgindade	Treatment of gynecological problems, diarrhea, and decubitus ulcers	[63]
*Stryphnodendron polyphyllum* Mart.	Barbatimão	Treatment of inflammation and infection, and used to promote healing	[77]
*Stryphnodendron rotundifolium* Mart.	Barbatimao	Treatment of leucorrhea and diarrhea; as an anti-inflammatory and antiseptic agent; and used to promote blood clotting and wound healing	[77]
*Tachigali aurea* Tul.	N/F	Treatment of scabies and used as an antimalarial agent	[53]
*Vatairea macrocarpa* (Benth.) Ducke	Amargoso, maleiteira and Angelim-do-Cerrado	Treatment of diabetes	[78]
*Zornia brasiliensis* Vogel	Urinária, urinana, and carrapicho	Diuretic agent and used for treatment of venereal diseases	[76]
**Lamiaceae**			
*Hyptis crenata* Pohl ex Benth.	Hortelã-brava or hortelã do campo	Treatment of gastrointestinal disturbances, including gastric ulcers	[79]
*Hyptis passerina* Mart. ex Benth.	N/F	N/F	[22]
*Hyptis radicans* (Pohl) Harley & J.F.B. Pastore	N/F	N/F	[22]
**Lauraceae**			
*Aiouea trinervis* Meisn	N/F	N/F	[22]
*Nectandra amazonum* Ness	Jigua or Canelo or Louro	N/F	[80]
*Nectandra gardneri* Meisn.	N/F	N/F	[22]
*Nectandra hihua* (Ruiz & Pav.) Rohwer	N/F	N/F	[22]
*Nectandra lanceolata* Nees	N/F	N/F	[22]
*Nectandra megapotamica* (Spreng.) Mez	Canela-lora, canela-preta or canela-do-mato	Treatment of rheumatism and pain	[81]
*Ocotea lancifolia* (Schott) Mez	Canela pilosa and laurel né	N/F	[82]
*Ocotea velloziana* (Meisn.) Mez	N/F	N/F	[22]
**Loganiaceae**			
*Strychnos pseudoquina* St. Hil.	Quina-quina	Treatment of digestive problems, anemia, diabetes, coughs, and headaches, and used as a vermifuge, depurative, and appetite-stimulating agent	[83]
**Lythraceae**			
*Lafoensia pacari* A.St.-Hil.	Mangava-brava, pacari, dedaleiro, louro-da-serra	Treatment of inflammatory conditions, gastric ulcers, wounds, fevers, and various types of cancer	[84]
**Malpighiaceae**			
*Banisteriopsis argyrophylla* (A. Juss.) B. Gates	Cipo-prata or cipó-folha-de-prata	Treatment of renal problems and used as an anti-inflammatory agent	[85]
*Byrsonima coccolobifolia* Kunth	Murici de flor rósea, murici-do-cerrado	Treatment of diarrhea	[63]
*Byrsonima crassa* A.Juss.	Murici-cascudo or Murici-vermelho	Treatment of snake bites, febrile illnesses, skin infections, diarrhea, and gastric disorders	[86]
*Byrsonima intermedia* A. Juss.	Murici-pequeno	Treatment of fevers, skin infections, stomach pain, diarrhea, and dysentery, and used as a diuretic and anti-asthmatic agent	[87]
*Byrsonima verbascifolia* (L.) Richard	Murici de flor amarela, murici-cascudo	Treatment of fever and diarrhea, and used as an astringent and mild laxative agent	[63]
**Malvaceae**			
*Guazuma ulmifolia* Lam	Mutamba, Chicomagro	Treatment of skin diseases and gastric ulcers	[88]
**Melastomataceae**			
*Miconia albicans* (SW.) Triana	Canela-develho	Treatment of rheumatoid arthritis, pain, and inflammation	[89]
*Mouriri elliptica* Martius	Puçá-preto or jaboticaba-do-cerrado, coroa-de-frade or coroa	Treatment of gastric ulcers and gastritis	[90]
*Mouriri pusa* Gardner	Pucá-preto, jaboticaba-do-cerrado	Treatment of gastric ulcers	[91]
*Pleroma stenocarpum* (Schrank et Mart. Ex DC.) Triana	N/F	N/F	[22]
**Meliaceae**			
*Cabralea canjerana* (Vell.) Mart.	Canjarana	N/F	[92]
*Guarea guidonia* (L.) Sleumer	Açafroa	Astringent, purgative, febrifuge, abortive, emetic, and anti-inflammatory agent	[26]
*Guarea kunthiana* A.Juss.	Jatuaúba	Antimalaric agent and used for treatment of stomach aches	[26]
**Metteniusaceae**			
*Emmotum nitens* Miers	Unha-d’anta, unha-de-anta	Treatment of hemorrhoids	[93]
**Moraceae**			
*Brosimum gaudichaudii* Trécul.	Inharé, mamacachorro, mamacadela	Treatment of infections, venereal diseases, furuncles, “impingem” (superficial skin mycoses), cancer, anemia, pneumonia, prickly heat, vitiligo, joint pain, inflammation, rheumatism, kidney diseases, and wounds, and used as a depurative and heart tonic agent	[93]
**Myristicaceae**			
*Virola sebifera* L.	Ucuúba-do-cerrado or mucuíba or Ucuúba, ucuúba branca-de-folha grande	Treatment of wounds and rheumatism	[18]
**Myrtaceae**			
*Blepharocalyx salicifolius* (Kunth) O.Berg	Murta	Treatment of respiratory diseases, coughs, colds, hypotension, rheumatism, hypoglycemia, diarrhea, leukorrhea, urethritis, and bladder diseases	[94]
*Campomanesia adamantium* (Cambess.) O. Berg	Gabiroba or guabiroba-do-campo or guavira	Antirheumatic, antidiarrheal, hypocholesterolemic, and anti-inflammatory, and used for treatment of cystitis and urethritis	[95]
*Campomanesia sessiliflora* (O.Berg) Mattos	N/F	N/F	[22]
*Campomanesia velutina* (Cambess) O. Berg	Gabiroba, guavira, cambuci	Treatment of diarrhea and intestinal cramps	[93]
*Eugenia dysenterica* (Mart.) DC.	Cagaiteira, cagaita	Purgative agent for treatment of diarrhea	[63]
*Eugenia involucrata* DC.	Pitanga vermelha or cereja pitanga do cerrado	Hypotensive, diuretic, antimicrobial, hypoglycemiant, and anti-inflammatory agent	[96]
*Eugenia klotzschiana* O.Berg	Pêra-do-cerrado, Cabacinha	N/F	[97]
*Eugenia uniflora* L.	Pitanga or pitangueira	Treatment of intestinal disorders and hypertension	[98]
*Myrcia bella* Cambess	Mercurinho	Treatment of gastrointestinal disorders and both hemorrhagic and infectious diseases	[99]
*Myrcia linearifolia* Cambess	N/F	N/F	[99]
*Myrcia splendens* (Sw.) DC.	N/F	Treatment of gastrointestinal disorders and both hemorrhagic and infectious diseases	[99]
*Myrcia variabilis* Mart. ex DC.	N/F	N/F	[22]
*Psidium brownianum* Mart. ex DC	Araçá-de-veado, murtinha do mato	Treatment of influenza and fever	[100]
*Psidium guineense* Sw	Goiabinha-araçá, araçá-do-campo, araçá verdadeiro or goiabinha selvagem	Treatment of inflammation and gastrointestinal disorders, and used as a diuretic agent	[101]
*Psidium laruotteanum* Cambess.	Araçá-Cascudo	N/F	[22]
*Psidium myrsinites* DC	Araçá	Treatment of cicatrization and diarrhea	[22]
*Psidium cattleyanum* Sabine	araça-rosa, araça-vermelho, or araça do campo	Adstringent, hepatoprotective, antidiarrheal, and analgesic agent	[102]
**Nyctaginaceae**			
*Neea theifera* Oerst.	N/F	N/F	[22]
**Ochnaceae**			
*Ouratea castaneifolia* (DC.) Engl.	Farinha-seca or mangue-do-mato or Tuiohy	Tonic and astringent agent	[20]
*Ouratea semiserrata* (Mart. & Nees) Engl.	N/F	N/F	[22]
*Ouratea spectabilis* (Mart.) Engl.	Folha-de-serra or batiputá	Treatment of diseases of the liver and skin	[103]
**Phytolaccaceae**			
*Gallesia integrifolia* (Spreng.) Harms	Pau-d’alho or garlic plant	Treatment of microbial, respiratory, and skin infections	[104]
**Piperaceae**			
*Piper aduncum* L.	Matico	Anti-inflammatory and antiseptic agent for the promotion of wound healing and for treatment of rheumatic conditions and diarrhea	[105]
**Polygonaceae**			
*Polygonum spectabile* Mart.	Erva-de-bicho	Stimulant and anti-helminths agent, and for treatment of hemorrhoids, diarrhea, ulcers, and gingivitis	[46]
**Primulaceae**			
*Myrsine guianensis* (Aubl.) Kuntze	Caapororoca, capororoca and pororoca	Antiseptic, antiparasitic, and contraceptive agent	[106]
**Proteaceae**			
*Roupala montana* var. *brasiliensis* (Klotzsch) K.S.Edwards	Carne-de-vaca, Bosta-de-urubu	Treatment of intestinal and non-specific blood disorders	[66]
**Rubiaceae**			
*Genipa americana* L.	Jenipapo	Treatment of bronchitis, diabetes, and kidney disease	[27]
*Psychotria deflexa* DC.	N/F	N/F	[22]
*Psychotria prunifolia* (Kunth) Steyerm.	N/F	N/F	[22]
*Palicourea rigida* Kunth	Gritadeira, bate caixa and douradão	Antifungal, diuretic, hypotensive, antiulcerogenic, cicatrizing, and anti-inflammatory agent, and for treatment of coughs, stomach aches, and kidney pains	[107]
*Psychotria capitata* Ruiz & Pav.	N/F	N/F	[22]
*Psychotria hoffmannseggiana* (Willd. ex Schult.) Müll.Arg.	N/F	N/F	[22]
**Rutaceae**			
*Spiranthera odoratissima* A. St.-Hi	manacá	Blood purgative and appetite-stimulating agent, and for treatment of renal and hepatic diseases, stomach aches, headaches, sore muscles, hepatic dysfunction, and rheumatism	[108]
*Zanthoxylum rhoifolium* (Lam.)	Mamica de cadela, mamica de porca	Roots are used as a febrifuge, digestant, and tonic; stem bark is used to treat flatulence, colic, dyspepsia, earaches, toothaches, and snake bites	[109]
*Zanthoxylum riedelianum* (Engl.)	Laranjeira-Brava, Limãozinho Branco, Mamonilha-De-Porca, Mamicão, Mama-De-Porca, Tamanquaré, Limãozinho	Analgesic agent for treatment of toothaches, inflammation, rheumatism, and skin stains	[110]
**Salicaceae**			
*Casearia sylvestris* Sw. var. sylvestris	Guaçatonga	Anti-inflammatory and anti-spasmodic agent, and for treatment of diarrhea, leprosy, fever, syphilis, herpes, and snake bites	[111]
**Sapindaceae**			
*Cupania cinerea* Poepp. & Endl.	N/F	N/F	[22]
*Cupania vernalis* Cambess.	Arco-de-barril, rabo-de-bugio	Treatment of inflammation and used as a febrifugic agent and tonic	[112]
*Matayba guianensis* Aubl.	Camboatá	N/F	[25]
*Serjania lethalis* A.St.-Hil.	Cipó-timbó, timbó	Piscicidal, used topically to treat pain	[19]
*Serjania marginata* Casar.	Cipó-uva or cipó-timbó	Treatment of gastric pain	[113]
**Sapotaceae**			
*Pouteria ramiflora* (Mart.) Radlk.	Curriola (curiola), brasa-viva, figo-do-cerrado, grao-de-galo, fruta-do-veado, massaranduba or maçaranduba, pessegueiro-do-cerrado, abiu-cutite, and pitomba-de-leite	Antihyperlipidemic agent and for treatment of worms, dysentery, pain, and inflammation	[114]
*Pouteria torta* (Mart.) Radlk	Guapeva, curiola, acá ferro, abiu do cerrado, and grão de galo	Antidysenteric	[115]
**Simaroubaceae**			
*Simarouba versicolor* A. St.-Hil.	Mata-barata	Insecticide, vermifuge, febrifuge, and antisyphilitic agent	[26]
**Siparunaceae**			
*Siparuna guianensis* Aubl.	Folha-santa, Negramina, Mõe-Hanakë, Limão-Bravo, Caápitiú, Capitiú	Tarminative, aromatic, stimulant, antidispeptic, and diuretic agent, and for treatment of back pain, rheumatism, and arthritis	[116]
*Smilax brasiliensis* Sprengel	Salsaparrilha or japecanga	Diuretic, diaphoretic, stimulant, anti-hypertensive, and antisyphilitic agent, and for treatment of arthritis, rheumatism, and skin disorders	[117]
**Solanaceae**			
*Solanum lycocarpum* A. St.-Hil.	Lobeira or fruta-do-lobo	Treatment of diabetes, obesity, and hypercholesterolemia	[118]
*Solanum palinacanthum* Dunal	Joá	Treatment of skin diseases	[119]
**Styracaceae**			
*Styrax camporum* Pohl	Laranjeira-do-mato	N/F	[22]
*Styrax ferrugineus* Nees & Mart.	Laranjinha do campo	Treatment of gastrointestinal diseases and fevers	[120]
**Verbenaceae**			
*Lippia lupulina* Cham.	N/F	Treatment of oral and throat infections	[121]
*Lippia origanoides* Kunth.	Salva-deMarajo and alecrim d’Angola	General antiseptic agent for the mouth, throat, and wounds, and for treatment of infant colic, diarrhea, indigestion, flatulence, heartburn, nausea, vaginal discharges, menstrual complaints, and fever	[122]
*Lippia salviaefolia* Cham.	N/F	N/F	[22]
**Vitaceae**			
*Cissus erosa* Rich.	Cipó-fogo	Treatment of warts and external ulcers	[20]
**Vochysiaceae**			
*Qualea grandiflora* Mart.	Pau-terra	Treatment of diarrhea and pain	[27]
*Qualea multiflora* Mart.	N/F	Treatment of external ulcers, gastric diseases, and inflammation	[123]
*Qualea parviflora* Mart.	Pau-terra, pau-ferro, pau-de-tucano	Treatment of diarrhea, blood diseases, intestinal colic, amebiasis, skin diseases, and inflammation, specifically ulcers and gastritis	[124]

N/F: Not Found.

## Data Availability

Not applicable.

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
