# Peer review of "Toxic Potential of Cerrado Plants on Different Organisms"

_ijms, 2022, doi:10.3390/ijms23073413_

Round 1

Reviewer 1 Report

In the presented form, the article should be qualified as a short communication, not Review ! There are 9 authors for such a short article. Please write  what each author did.

In the introduction, I miss information about Cerado:

- what is unusual in this biome,
- what plant families dominate,
- how many endemics there are.

The article lacks ethnobotanical issues. Do the local people use Cerado plants?

Please add two minimum two plates showing the most important Cerado plants that have potential in medicine and pharmacology.

Reviewer 2 Report

The review "Toxic potential of Cerrado plants on different organisms" fits the journal's scope. The authors reviewed the literature regarding the toxicity of plants from Cerrado. The review is the result of a very organized team work, and present general data regarding the plant species names, plant systematic, traditional use and scientific data regarding some biological activities. The results are presented clearly, and supplemented by discussions. The supplementary material provide an impressive amount of data whch would be very helpful to other researchers. Although the authors put a lot of efforts in this work, some points should be clarified:

  1. The aim of the work is not clearly stated. For example, it is not very clear if the toxicity "on different organisms" is always related with toxicity on humans. The objective seems to be the indication of some species which can be used to obtain some medicines that can be safely handled and administered. However, given the high amount of plant species, and corroborated with a greater number of metabolites, this scope is practically impossible to achieve. The biodiversity highlighting seems a more proper aim of this study, and the authors should develop this direction.
  2. No significant data are provided regarding the Cerrado region. Please add information regarding the climate, endemic species, and what other data you consider appropriate to sustain the phrase from lines 54-55.          

Other comments

lines 112-114 - please rephrase. The phrase is too general and is not proper

lines 312-315 - please rephrase - especially "that can be safely handled and administered"

please use italics for plant names

please increase the quality of fig. 4 and 6

line 283 - please correct the error

table S2 - please resize the table

Round 2

Reviewer 1 Report

Authors improved manuscript but some other changes are required.  The "Table 1S Ethnobotanical data for the Cerrado biome plant species included in the present review" is important and should be included in the main text. Also authors should provide two plates with photos of the main important Cerado plant species which are used medicine and pharmacology.
